# ATM-Mediated Mitochondrial Radiation Responses of Human Fibroblasts

**DOI:** 10.3390/genes12071015

**Published:** 2021-06-30

**Authors:** Tsutomu Shimura

**Affiliations:** Department of Environmental Health, National Institute of Public Health 2-3-6 Minami, Wako 351-0197, Saitama, Japan; simura.t.aa@niph.go.jp; Tel.: +81-48-458-6261

**Keywords:** ATM, mitochondria, radiation, ROS, oxidative stress

## Abstract

Ataxia telangiectasia (AT) is characterized by extreme sensitivity to ionizing radiation. The gene mutated in AT, Ataxia Telangiectasia Mutated (ATM), has serine/threonine protein kinase activity and mediates the activation of multiple signal transduction pathways involved in the processing of DNA double-strand breaks. Reactive oxygen species (ROS) created as a byproduct of the mitochondria’s oxidative phosphorylation (OXPHOS) has been proposed to be the source of intracellular ROS. Mitochondria are uniquely vulnerable to ROS because they are the sites of ROS generation. ROS-induced mitochondrial mutations lead to impaired mitochondrial respiration and further increase the likelihood of ROS generation, establishing a vicious cycle of further ROS production and mitochondrial damage. AT patients and ATM-deficient mice display intrinsic mitochondrial dysfunction and exhibit constitutive elevations in ROS levels. ATM plays a critical role in maintaining cellular redox homeostasis. However, the precise mechanism of ATM-mediated mitochondrial antioxidants remains unclear. The aim of this review paper is to introduce our current research surrounding the role of ATM on maintaining cellular redox control in human fibroblasts. ATM-mediated signal transduction is important in the mitochondrial radiation response. Perturbation of mitochondrial redox control elevates ROS which are key mediators in the development of cancer by many mechanisms, including ROS-mediated genomic instability, tumor microenvironment formation, and chronic inflammation.

## 1. Introduction

Ataxia telangiectasia (AT) is a rare autosomal recessive disorder characterized by progressive impairments in muscular coordination, immunodeficiency, radiosensitivity, and a predisposition to cancer. The overall prevalence of AT is about 1:100,000 live births in Japan. The gene that is mutated in AT, Ataxia Telangiectasia Mutated (ATM) and ATM- and Rad3-related (ATR), encodes large serine/threonine protein kinases that orchestrate nuclear DNA damage responses (DDR) with a multitude of substrates. ATM and ATR have distinct DNA damage specificities, and their functions are not identical. Double-strand breaks (DSB) can arise from metabolic processes, DNA replication stress, and DNA-damaging agents such as ionizing radiation, UV light, and genotoxic molecules. Recognizing DNA damage is the critical first step of the DDR. The trimeric MRE11–RAD50–NBS1 (MRN) complex is involved in DSB recognition, pre-repair mechanisms, and keeping sister chromatids or broken ends in close proximity to one another [1]. This complex recruits ATM to the DSB site [2], which, in turn, initiates DNA repair and checkpoint responses by activating an extensive signaling network [3,4,5,6]. ATM is involved in maintaining genomic integrity and defends against endogenous as well as exogenous DNA damage [7]. ATM also has a role as a sensor of ROS and is activated by treatment with H_2_O_2_ [8]. Loss of ATM function leads to nuclear genomic instability and reactive oxygen species (ROS)-mediated oxidative stress, which is thought to be a causal factor in the development of AT [5,6,7,9,10,11]. Elevated oxidative stress and mitochondrial abnormalities have been reported in AT patients and in ATM-deficient mice [10,12,13,14]. Antioxidants can prevent intrinsic defects and cancer susceptibility in ATM-deficient mice [15,16]. However, how ATM is involved in cellular redox homeostasis remains unclear. This article will review current research regarding the role of ATM in radiation-induced oxidative stress, in which the mitochondria play a critical role. Mitochondrial ROS are associated with radiation-related cancers. An analysis of radiation-induced oxidative stress is vital to estimate radiation cancer risks.

## 2. Mitochondrial ROS Generation and Redox Control

ROS are highly reactive chemical ions such as superoxide anions (O_2_^−^), hydroxyl radical (OH^−^), peroxy radical (RO_2_^−^), hydrogen peroxide (H_2_O_2_), and reactive nitrogen species. Mitochondria are central to energy production, generating adenosine triphosphate (ATP) via oxidative phosphorylation (OXPHOS). Endogenous ROS are generated during necessary metabolic reactions in the mitochondria of eukaryotic cells. Mitochondrial DNA (mtDNA) is located near the site of ROS production (Lambert and Brand, 2009) and is therefore especially vulnerable to ROS-induced damage [17]. In turn, this makes genes encoded by mtDNA especially sensitive to oxidative-stress-induced mutations (Wallace, 2010). Complexes I and III in the mitochondria are the sites of ROS production and electron leak from the electron transport chain reaction in the form of O_2_^−^ [18,19,20]. The nicotinamide adenine dinucleotide phosphate oxidative enzymes (NOXs) produce ROS in phagocytes and in other tissues [21]. ROS attack a large number of biomolecules such as proteins, lipids, carbohydrates, and nucleic acids. H_2_O_2_ is less reactive than O_2_^−^; it can also diffuse across the mitochondria to mediate oxidative signaling [22].

The cellular redox balance is regulated by eliminating ROS with antioxidants. Molecules involved in the mitochondrial antioxidant defense system such as glutathione (GSH) are required for maintaining intracellular redox homeostasis [23]. Manganese superoxide dismutase (MnSOD) and glutathione peroxidase (GPx) are responsible for removing ROS. MnSOD catalyzes the dismutation of O_2_^−^ into H_2_O_2_ and O_2_. GPx is a scavenger of H_2_O_2_ that converts GSH into oxidized glutathione (glutathione disulfide, GSSG) and simultaneously reduces H_2_O_2_ into H_2_O [23,24,25]. GSH is regenerated from GSSG using NADPH-dependent glutathione reductase. The GSH/GSSG ratio is thought to be a useful indicator of oxidative stress within cells. Further, disruption of cellular redox homeostasis leads to ROS-mediated oxidative stress which causes detrimental health effects, including cancer, neurodegenerative diseases, and cardiac diseases [26,27,28,29,30].

## 3. Radiation-Induced Oxidative Stress

Radiation induces oxidative stress in cells [31]. Radiation primarily generates ROS via the radiolysis of water. Subsequently, the delayed production of mitochondrial ROS mediates the long-lasting effects of radiation [32]. Cellular antioxidant defense systems maintain ROS levels after exposure to low-to-moderate doses of ionizing radiation. Acute single radiation (SR) of less than 2 Gy transiently increases ROS concentrations, which subsequently return to baseline levels 24 h after irradiation, in human fibroblasts [32,33]. In contrast, high doses of radiation (more than 5 Gy) are associated with the induction of oxidative stress, suggesting that the antioxidant defense system is ineffective in this scenario. The GSH reaction is critical in maintaining redox homeostasis following radiation. The effect of radiation on GSH redox control was investigated in human fibroblasts (manuscript submitted). Although reduced GSH was present in the irradiated cells, radiation inactivated GPx. Different radiation exposure methods were used to evaluate the effects of fractionated radiation (FR) on radiation-induced oxidative stress. Clinical radiotherapy is given by dividing a dose of radiation into multiple fractions over several weeks [34]. Repeated radiation exposures in low doses (0.01 Gy or 0.05 Gy per fraction) for 21 days (total dose of 0.3 Gy or 1.5 Gy) induced prolonged ROS generation, leading to chronic oxidative stress in FR-treated cells [35]. Low doses of FR are equivalent to high doses of SR on mitochondrial ROS induction. Similarly to SR, FR in low doses causes downregulation of GPx activity. Thus, a decrease in GPx activity is an important contributor to radiation-induced oxidative stress. Essentially, radiation increases mitochondrial ROS levels according to the radiation dose and irradiation conditions in human fibroblasts [26].

## 4. Mitochondrial Radiation Responses

The role of mitochondria in the radiation response is well summarized in other mitochondrial biology papers [36]. The effects of ionizing radiation on the nuclei and mitochondria are illustrated in Figure 1. Radiation increases mitochondrial ROS levels according to the radiation dose and irradiation conditions in human fibroblasts [32]. Radiation-induced oxidative damage of mtDNA can be detected as increases in 8-hydroxydeoxyguanosine (8-OHdG) levels via high-performance liquid chromatography–electrochemical detectors (HPLC–ECD) [35]. Radiation activates mitochondrial biogenesis to aid in the increased energy demands for DDR [35,37]. As described above, mitochondria have a role in ROS generation during OXPHOS under irradiation. Radiation damage to the mitochondria leads to impairments in mitochondrial respiration and makes them more prone to ROS generation [36,38]. This leads to a vicious cycle of further ROS production and mitochondrial DNA damage. The selective removal of damaged mitochondria, known as mitophagy, is indispensable to cellular survival in cases where damaged mitochondria are releasing cytotoxic materials. In response to a loss of mitochondrial membrane potential, phosphatase and tensin homolog induced putative kinase 1 (PINK1), a mitochondrial kinase, phosphorylates Parkin, an E3-like ubiquitin-ligase, which, in turn, localizes specifically to impaired mitochondria [39]. Radiation-induced mitochondrial ROS are associated with the induction of mitophagy, which can be detected by the formation of Parkin and activation of the antioxidative response of nuclear factor erythroid 2-related factor 2 (Nrf2) in human fibroblasts [32,35]. Parkin can be utilized as a marker to identify oxidative damage in mitochondria.

Mitochondria undergo morphological changes in order to maintain healthy mitochondrial networks by a balance of fission and fusion events [40,41]. Following radiation exposure, the mitochondrial morphology is remarkably altered to maintain its function. Such mitochondrial dynamics occur via the fusion of healthy mitochondria with the damaged ones [42,43,44]. In fact, low-dose radiation stimulates mitochondrial fusion in order to protect rat neurons [45]. On the other hand, a high dose of radiation induces dynamin-related protein 1 (Drp1)-mediated mitochondrial fission in normal human fibroblast-like cells [46]. Mitochondria are also associated with radiation-induced adaptive responses, the mitochondrial bystander effect, and genomic instability. MnSOD exists within mitochondria and promotes adaptive radiation responses as a signaling regulator of cell survival pathways [47]. ROS activates NF-kappaB signaling pathways, which are implicated in the regulation of the radiation-induced bystander effect [48]. Intercellular communication is mediated by mitochondrial transfer from one cell to another under stress conditions [49]. Mitochondria are key players in activating apoptosis in mammalian cells. In apoptosis, cytochrome c is released from the mitochondria by the permeability transition pore and subsequently activates apoptosis signaling pathways. [50]. Collectively, mitochondria are an important radiation target.

A part of ATM is located in the mitochondria and responds to mitochondrial dysfunction [51]. Radiation activates ATM, which protects mitochondrial quality [32,52,53]. ATM has a role in inducing mitophagy after selective elimination of damaged mitochondrial components by mitochondrial fission. Additionally, ATM loss leads to severe radiation-induced mitochondrial damage in human fibroblasts [52]. ATM is frequently mutated in cancers. Cells with ATM mutation may display a highly radiosensitive phenotype with a lack of the normal ATM-mediated mitochondrial damage response including OXPHOS activation, ROS generation, and mitophagy. ATM plays a direct role in modulating mitochondrial homeostasis under genotoxic stress.

## 5. ATM and AMPK-Mediated Interconnected Damage Signaling Networks between the Nuclei and Mitochondria

Radiation immediately stimulates multiple comprehensive molecular signaling pathways to execute DDR. ATM is considered the master sensor of DSBs and transduces the DNA damage signal to trigger the activity of downstream effectors by biochemical modification reactions [54]. The nuclear DNA damage signal diffuses to other intracellular organelles, including mitochondria, in irradiated cells. AMP-activated protein kinase (AMPK) protects cells against physiological stress by sensing energy stress when the ATP:AMP/ADP ratio declines and suppresses cell growth via regulating the mammalian target of rapamycin (mTOR) pathway [55]. AMPK controls various aspects of mitochondrial homeostasis by promoting mitochondrial biogenesis, regulation of mitochondrial morphology, and mitophagy [56]. One study suggests that ATM is an upstream kinase for the activation of AMPK by phosphorylation on Thr172 [57], while another study suggests that ATM does not directly phosphorylate AMPK [58]. The precise mechanism of ATM-mediated phosphorylation of AMPK after irradiation is under current investigation. ATM controls mitochondrial quality following radiation exposure [32,52,53]. AMPK acts as a mediator of DNA damage signals to mitochondria in response to genomic stress. DNA damage activates the ATM–AMPK–peroxisome proliferator-activated receptor **γ** coactivator 1α (PGC1α) signaling pathway, which facilitates mitochondrial biogenesis (Figure 2) [52,57]. ROS released from mitochondria cause indirect oxidative damage to nuclear DNA, which subsequently induces long-lasting oxidative damage, in irradiated cells. AMPK activates p53 and cyclin-dependent kinase inhibitors p21-dependent cell cycle arrest [59]. Collectively, ATM and AMPK are key molecules in the crosstalk between the nucleus and mitochondria under genomic stress conditions inflicted by ionizing radiation.

## 6. The Role of Oxidative Stress in Cancer

High ROS concentrations in cancerous cells have been proposed to arise from mitochondrial dysfunction [60]. Mitochondrial genomic instability is associated with increased incidence of various diseases, including metabolic diseases, neurodegenerative diseases, and cancer [27,61,62,63]. Oxidants have genotoxic effects and can promote the development of multistage carcinogenesis [64]. Indeed, mtDNA mutations have been reported in various types of tumors [65,66,67]. Impairment of mitochondrial respiration alters the cellular metabolism of cancer cells [68]. Cancer cells require a much higher glucose supply to maintain their high proliferation rate, and they prefer to metabolize glucose by glycolysis. The mitochondrial antioxidant enzyme MnSOD contributes to the protection of mitochondria against oxidative stress and plays a role in tumor prevention [69]. These results indicate a possible association between mitochondria-mediated oxidative stress and carcinogenesis [36,70]. ROS are key mediators in the development of cancer via many mechanisms, including ROS-mediated genomic instability, tumor cell proliferation, and chronic inflammation [71,72].

The tumor microenvironment has been widely implicated in tumorigenesis. Tumor microenvironments consist of stromal cells including myofibroblasts and/or cancer-associated fibroblasts (CAFs), vascular cells, and immune cells. Stromal fibroblasts in cancer, called CAF, communicate with malignant tumor cells via the release of tumor cell growth factors and play key roles in tumor initiation, progression, and metastasis. α-smooth muscle actin (α-SMA) is a known marker of CAF [73]. ROS-mediated signaling can regulate the formation of tumor microenvironments in tumorigenesis [74]. We recently revealed that radiation affects malignant cancer cells and can also cause molecular alterations in stromal fibroblasts. Radiation stimulates fibroblast activation via mitochondrial ROS-mediated transforming growth factor-β signaling [75]. Interactions between radiation-activated fibroblasts and malignant cancer cells contribute to the formation of the tumor microenvironment [75]. Mitochondrial dysfunction may therefore facilitate radiation-induced carcinogenesis.

## 7. Conclusions

ATM has a critical role in mitochondrial radiation responses. Cross-talk between the nuclei and mitochondria following radiation is important in maintaining cellular redox control. Mitochondrial oxidative stress is a key player in radiation-induced cancers. Further investigation is needed to clarify the risks of radiation.

## Figures and Tables

**Figure 1 genes-12-01015-f001:**
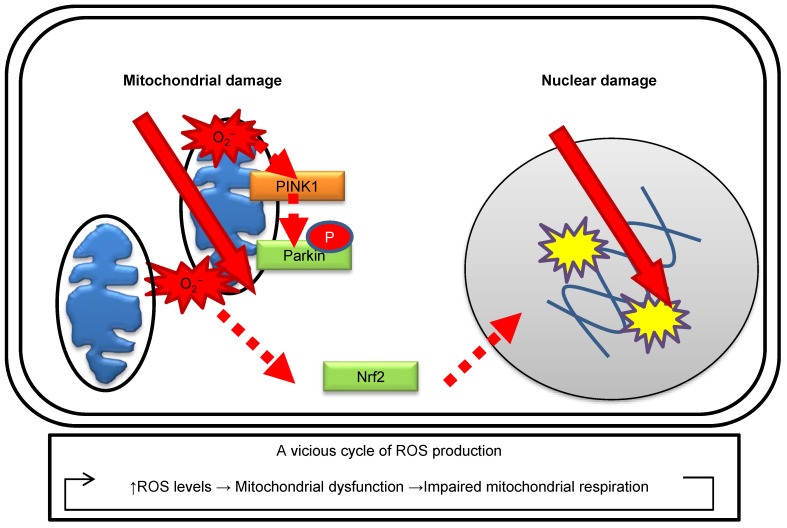
Effects of ionizing radiation on the nuclei and mitochondria.

**Figure 2 genes-12-01015-f002:**
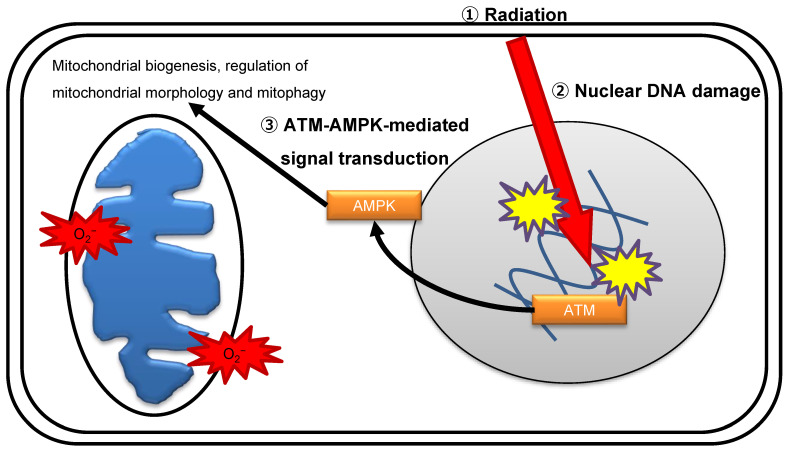
ATM-mediated crosstalk between the nuclei and mitochondria.

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
