# Peer review of "ATM-Mediated Mitochondrial Radiation Responses of Human Fibroblasts"

_genes, 2021, doi:10.3390/genes12071015_

Round 1

Reviewer 1 Report

An interesting review, well written and organized. 

Here some specific comments: This review discusses ATM-mediated mitochondrial radiation responses of human fibroblasts. The aim of the review is interesting, focusing on the role of mitochondria in ROS production and redox control, radiation-induced oxidative stress and its role in cancer.  Conclusions are consistent with the evidence and arguments presented, with further investigations needed to understand and clarify the risk of radiations, as they say. The text is easy to read, clear and well organized.

Author Response

Point-by-point answers to the comments:

We have marked all the changes in the revised manuscript with red. Following are our reply and changes in response to the comments and suggestions.

Reviewer #1:

An interesting review, well written and organized.

Here some specific comments: This review discusses ATM-mediated mitochondrial radiation responses of human fibroblasts. The aim of the review is interesting, focusing on the role of mitochondria in ROS production and redox control, radiation-induced oxidative stress and its role in cancer.  Conclusions are consistent with the evidence and arguments presented, with further investigations needed to understand and clarify the risk of radiations, as they say. The text is easy to read, clear and well organized.

We greatly appreciate editors and reviewers for spending their precious time. 

Reviewer 2 Report

This article reviewed the role of ATM kinase on the radiation response of mitochondrion. I believe it will be a useful reference for readers because of its concise description.

Major comments

  1. Section ”4 mitochondrial radiation response” may need an illustration to understand.
  2. Line 154: It is well known relatively high dose of radiation induce the fission of mitochondria. Does ATM affect the fission of mitochondria too? The author should add the effect of radiation on mitochondria dynamics both fission and fusion as background information.

Minor comments

  1. If possible, it is better to describe the difference between ATM and ATR functions briefly.
  2. Line 143, radiation dose should be written. as the same as line 104?
  3. Line 122. ROS could not accumulate, maybe better to use another phrase such as oxidative stress, etc.

Author Response

Point-by-point answers to the comments:

We have marked all the changes in the revised manuscript with red. Following are our reply and changes in response to the comments and suggestions.

Reviewer #2:

This article reviewed the role of ATM kinase on the radiation response of mitochondrion. I believe it will be a useful reference for readers because of its concise description.

Major comments

Section ”4 mitochondrial radiation response” may need an illustration to understand.

Answer: I would like to thank the reviewer for his/her valuable comments on the manuscript. Mitochondrial radiation response is illustrated in Figure 1 in the revised manuscript.

Line 154: It is well known relatively high dose of radiation induce the fission of mitochondria. Does ATM affect the fission of mitochondria too?

The author should add the effect of radiation on mitochondria dynamics both fission and fusion as background information.

Answer: Mitochondria undergo morphological changes in maintaining functional mitochondria by a balance of fission and fusion events. In fact, low-dose radiation stimulates mitochondrial fusion in order to protect rat neurons (Chien et al., 2015). On the other hand, high dose of radiation induces dynamin-related protein 1 (Drp1)-mediated mitochondrial fission in normal human fibroblast like cells. ATM has a role on inducing mitophagy after selective elimination of damaged mitochondrial components by mitochondrial fission. We described this in the revised manuscript.

Minor comments

If possible, it is better to describe the difference between ATM and ATR functions briefly.

Answer: We described the difference between ATM and ATR functions briefly. The DNA damage specificities of ATM and ATR differ from each other, and their functions are not identical.

Line 143, radiation dose should be written. as the same as line 104?

Line 122. ROS could not accumulate, maybe better to use another phrase such as oxidative stress, etc.

Answer: We corrected.

We greatly appreciate editors and reviewers for spending their precious time. 

Reviewer 3 Report

This short review discusses cellular responses to radiation and the involvement of the protein kinase ATM in these processes.

Radiation-induced oxidative stress is well-covered here, as well as the antioxidant mechanisms that act to neutralize this stress. Treatment of ATM and its roles related to oxidation are not very well covered though. For instance, ATM is reported to be a sensor of ROS and its loss results in high levels of specific ROS species, which is not covered here. Also, it is not very clear from the text how the normal responses of ATM with DNA damage compare to the ATM responses to radiation which has a ROS component. Similarly, there have been many articles about roles of ATM in mitochondria, but are there differences in mitochondrial outcomes that are specifically seen with radiation exposure? Lastly, ATM is frequently mutated in cancers; there could be at least some discussion of this and whether it is likely that ROS levels or oxidative stress are relevant to cancer or to responses to radiation therapy.

Minor issues:

The title suggests that the subject here is specifically fibroblasts, but from the text that does not seem to be the case.

line 63: "therefore especially vulnerable to ROS-induced damaged"

Zhang et al investigated mitochondrial ROS in the context of ATM deficiency (PMID 29991649); this would be useful to cite here.

Author Response

Point-by-point answers to the comments:

 We appreciate editor and reviewers for spending their precious time. We have marked all the changes in the revised manuscript with red. Following are our reply and changes in response to the comments and suggestions.

Reviewer #3:

This short review discusses cellular responses to radiation and the involvement of the protein kinase ATM in these processes. Radiation-induced oxidative stress is well-covered here, as well as the antioxidant mechanisms that act to neutralize this stress. Treatment of ATM and its roles related to oxidation are not very well covered though. For instance, ATM is reported to be a sensor of ROS and its loss results in high levels of specific ROS species, which is not covered here.

Answer: We would like to thank the reviewer for his/her valuable comments on the manuscript. ATM also has a role as a sensor of ROS and is activated by treatment with H2O2 (Guo et al., 2010). Loss of ATM function leads to nuclear genomic instability and reactive oxygen species (ROS)-mediated oxidative stress which is thought to be a causal factor in the development of AT. We described this in the revised manuscript.

Also, it is not very clear from the text how the normal responses of ATM with DNA damage compare to the ATM responses to radiation which has a ROS component.

Answer: Radiation induces not only nuclear DNA damage response but also mitochondrial radiation responses (Figure 1). ATM is involved in signal transduction from nucleus to the mitochondria in response to ionizing radiation (Figure 2). Oxidative phosphorylation is activated in response to radiation-induced nuclear DNA damage, and this also generates mitochondrial ROS as by-product.

Similarly, there have been many articles about roles of ATM in mitochondria, but are there differences in mitochondrial outcomes that are specifically seen with radiation exposure?

Answer: As mentioned by the reviewer, many articles discusses about roles of ATM in mitochondrial function. However, the role of ATM on mitochondrial radiation responses is unclear. The aim of this review paper is to introduce our current research surrounding the role of ATM on maintaining cellular redox control following radiation. ATM signaling is essential in the mitochondrial radiation responses in human fibroblasts.

Lastly, ATM is frequently mutated in cancers; there could be at least some discussion of this and whether it is likely that ROS levels or oxidative stress are relevant to cancer or to responses to radiation therapy.

Answer: ATM loss eliminated the effect of radiation on mitochondria in human fibroblasts (Shimura et al., 2016a). Cancer cells with ATM mutation may display a highly radiosensitive phenotype with a lack of the normal ATM-mediated mitochondrial damage response. We described this in the revised manuscript.

Minor issues:

The title suggests that the subject here is specifically fibroblasts, but from the text that does not seem to be the case.

Answer: We are sorry for lack of proper explanation. The data we discussed in this review paper were obtained by using human fibroblasts.

line 63: "therefore especially vulnerable to ROS-induced damaged"

Zhang et al investigated mitochondrial ROS in the context of ATM deficiency (PMID 29991649); this would be useful to cite here.

Answer: Thanks the reviewer for providing me the information. We added above manuscript as a reference.

We greatly appreciate editors and reviewers for spending their precious time.